# Finger-palm synergistic soft gripper for dynamic capture via energy harvesting and dissipation

Yin Zhang [1], Wang Zhang[2], Pan Gao[1], Xiaoqing Zhong[1] & Wei Pu [1] ✉

Robotic grippers, inspired by human hands, show an extraordinary ability to manipulate objects of various shapes, sizes, or materials. However, capturing objects with varying kinetic energy remains challenging, regardless of the classical rigid-bodied or frontier soft-bodied grippers. Here, we demonstrate a rapid energy harvesting and dissipation mechanism for the soft grippers leveraging the finger-palm synergy. Theoretically and experimentally, this mechanism enables a soft gripper to reliably capture high-speed targets by dissipating and harvesting almost all the target's kinetic energy within 30 milliseconds. The energy harvesting and dissipating capability are adjustable and can be enhanced by inflating pressure. Additionally, the harvested energy is autonomously transferred into fingers to enhance their grasping force and reduce the response time. To highlight, the grippers we developed are integrated into a six-rotor drone and successfully capture flying objects in an outdoor experiment. These results significantly advance robotics development in achieving dynamic capture of dynamic targets.

Softness and body compliance are salient features frequently exploited by biological systems, aiming to seek simplicity and reduce complexity in their interactions against the environment[1]. Soft materials in robotics are a promising direction to build novel robotic systems with new capabilities that promise to interact more effectively with natural, unstructured environments and humans. Scientists in robotics-related fields draw inspiration from biological systems and seek to utilize soft materials such as silicone rubber[2,3], hydrogel[4], polyurethane[5,6], etc., to empower soft robotics with functions and properties similar to biological organisms or their muscles, such as grasping[7–9], swimming[10,11], growing[12,13], flying[14], climbing[15,16] and more. Low-modulus soft materials endow these soft machines with outstanding performance and functionality along with limitations in resisting external impacts, posing a challenge but also great potential for these soft devices to be applied to unstructured complex scenes[12,13,17].

Animals moving in the real world are buffeted by dynamic loads imposed by the environment[18]. Energy storage and exchange mechanisms found in many organisms play a great important role in keeping their mechanical stability after perturbations. For example, the elastic energy storage in the musculoskeletal system of polypods determines their operation dynamics[19–21]. Birds maintain their balance by leaning their bodies upward and stretching their legs and feet to absorb the bird's momentum while perching[22]. Commonly, dexterous human hands are capable of firmly grasping fast-flying basketballs or handballs through the cooperation of multiple tissues such as muscles, bones, and skin. In fact, achieving the above performance is very challenging if each component in a biological system function independently. On traditional hard-bodied robots (machines), some buffer devices, such as typical spring-link mechanisms[23], dampers[24], etc., are often used to absorb the energy of external interference loads. However, to the best of our knowledge, there is yet an efficient energy regulation mechanism in soft robotics design to resolve the energy of external impact loads.

Grasping has been extensively investigated in soft robotics, and effective grippers exist in manufacturing, prosthetics, and many other fields. Despite significant advances in robotic grippers, their performance in dynamic scenarios is still not comparable to biological grippers. For grasping static targets, existing grippers rely

[1]School of Aeronautics and Astronautics, Sichuan University, 610207 Chengdu, China. [2]Department of Mechanical Engineering, Massachusetts Institute of Technology, Cambridge, MA 02139, USA. ✉e-mail: Pwei@scu.edu.cn

on mechanical interlocking[25], frictional grip[26], and adhesion grip[17,27], which have enriched the envelope form between graspers and objects. Similarly, researchers using variable stiffness materials or structures[28,29], biomimetic adhesive sheets[17,30,31], and variable configuration designs[32] have made grippers compatible with objects of different shapes, masses, scales, and textures. However, for the dynamic targets, their large collision kinetic energy easily vulnerates both the grasped object and the rigid grasper vulnerable by causing irreversible destruction. If it is a soft gripper, the viscoelasticity of the soft material may be act as a booster for the object to escape. Overall, grasping dynamic targets faces two significant challenges: (1) systematical analysis of the collision/capture mechanism and (2) minimizing the impact energy between the dynamic target and the gripper.

In this paper, we present a finger-palm synergistic energy dissipation mechanism (FPSED) that enables a soft gripper to capture dynamic targets reliably. Part of the energy of the dynamic target is rapidly dissipated during a brief collision with the palm, and part of the energy is harvested into fingers for enhanced grip and less response time (Fig. 1a, d). This concept can be generalized to other machines and robots. Here we demonstrate its effectiveness through a tendon-driven, sequential motion soft gripper with a soft palm. First, we design an energy processing system based on the proposed FPSED mechanism, model and analyze the FPSED mechanism, and compare it with four conventional grippers designs to demonstrate the mechanism's superiority for energy dissipation and capture. Second, we experimentally validate the established collision model and demonstrate the excellent compatibility of the FPSED mechanism with multiple target object velocities. In combination, we develop a two-finger and a three-finger gripper, respectively, to demonstrate the potential of the soft gripper of the FPSED mechanism to capture dynamic targets in a laboratory and natural outdoor environments (Fig. 1e).

## Results

### Components and working principle of the gripper

The soft gripper developed in this study consists of multiple fingers and a soft palm, shown schematically in Fig. 1a. An actuation module is used to control the opening and closing of the fingers. To tighten (fingers open) and release (fingers closed) all finger tendons (Fig. 1b), a motor and a planetary gear mechanism are applied within the actuator module, and a motor-driven sun gear that meshes with the surrounding planetary gears is employed. For the two-finger and three-finger grippers used in the following study, the difference in their actuation modules is the number of planetary gears (Fig. 1b(i), (ii)). A hollow, brushless direct-current (DC) motor with an integrated driver is adopted to facilitate wirings, including pipes, signals, and power cables. Thus, all cables and pipes on the soft gripper can be routed through the motor shaft to the control system. Furthermore, to empower the gripper to capture in any direction, we fabricate a tendon-driven bellows-like flexible manipulator by 3D printing and connect it to the soft gripper with a rigid connector. Three tendons on the manipulator are arranged every 120° along its circumference, and three servo motors are used to control the retraction and release of the three tendons in a winding manner, respectively.

Each finger has two main components. One is a soft fiber-reinforced bending actuator (Fig. 1c(i)) made of silicone rubber with a shore hardness of 5 degrees. The manufacturing process is in the Methods section and Supplementary Fig. 1a. The Kevlar fiber constrains the expansion of the actuator along its radial direction. The inextensible layer prevents it from extending axially. When inflated ($\Delta P > 0$), the anisotropy of the actuator after being constrained is manifested as the actuator output bending moment and motion. Another component is a tendon-driven flexible skeleton (Fig. 1c(ii)). The skeleton is manufactured with thermoplastic polyurethane (TPU) rubber by 3D printing (Supplementary Fig. 1d). Referring to our previous research[33], the skeleton opens as the tendon tightens and

closes when it relaxes. Notably, only one tendon is required for a single skeleton, and three joints exhibit a sequential motion under the tension of one tendon when opening and closing based on the stiffness gradients at these joints. In the finger closure process, the end joint is closed first, followed by the middle and fingertip joint. During opening, the fingertip joint is opened first, followed by the middle and end joint (Supplementary Movie 6).

The sequential motion behavior enables the skeleton to wrap better and is compatible with objects of various shapes compared to the uniform motion of the fiber-reinforced soft actuator mentioned above. However, the fact that the skeleton goes from flat to closed depends only on the internal stress of the material, which results in limited grip force and a relatively long response time for the fingers. For the proposed gripper, we present a solution that considers both actuators' kinematic and mechanical properties, inserting the fiber-reinforced actuator into the flexible skeleton to form a composite (Fig. 1c(iii)). Thus, the finger's grasping force and response time depend on a combined function of the air pressure inside the soft actuator and the material stress stored in the deformed skeleton. Coordinated by the FPSED mechanism (detailed below), the pre-charged gas flows between the fingers and palm. The tendons of the fingers tighten ($F_{tendon}$ increases), the internal actuator volume decreases as the fingers open, and gas flows into the palm ($P_{finger}$ decreases). The tendon is released when grasping an object ($F_{tendon}$ decreases), the object squeezes the palm while the finger is closed, and the gas flows into the actuator of the finger ($P_{finger}$ increases).

### Finger-palm synergistic energy dissipation mechanism

Compared to static objects, the distinctive feature of dynamic objects is kinetic energy, which can be converted into elastic potential energy and accelerate their escape after colliding with soft elastic grippers (Supplementary Movie 1). Kinetic energy is proportional to the object's mass and the square of velocity. If the target velocity can be minimized quickly, then the collision energy could also be minimized. Therefore, we design a valve system to regulate the energy harvesting and dissipation efficiency in the FPSED mechanism. The valve system, working as a medium to coordinate energy exchange, is connected between the fingers and the palm. There are three working states of the valve system that correspond to the three essential processes of capturing dynamic targets: before collision, during collision, and capturing target. Before collision, the initial pressure in the palm and fingers is the same (Fig. 1d(i)). During collision, the kinetic energy of an object will be converted into gas internal energy to stiffen the gripper and improve gripping, and the gas in the palm and fingers can only flow in one direction (palm to fingers) (Fig. 1d(ii)). In detail, according to the ideal gas law, the volume of the palm $V_{palm}$ decreases after compression and deformation, and the internal air pressure $P_{palm}$ will increase rapidly. A portion of the gas in the palm will flow into two bending actuators of the gripper (the gas will not flow back to the palm because of the check valve) until the object's kinetic energy decays to zero, and the remaining gas will stay in the palm. At the end of the compression phase, the rapid release of the energy stored in the remaining gas of a collision would cause a target to bounce back quickly and escape. In order to solve this problem, we use a pressure sensor to detect whether the gas transfer has reached a limit value. Once the set threshold is triggered, the gripper starts to capture the target (Fig. 1d(iii)). Meanwhile, the control board quickly opens the exhaust valve. In this way, the energy stored in the palm is dissipated following rapid decompression. The kinetic energy of the target object is minimized after being harvested and rapidly dissipated so that the gripper has enough response time to capture.

### Collision dynamic model for the FPSED mechanism

To better understand the FPSED mechanism, we devise a dynamic model to analyze how the palm's damper rate and stiffness affect the

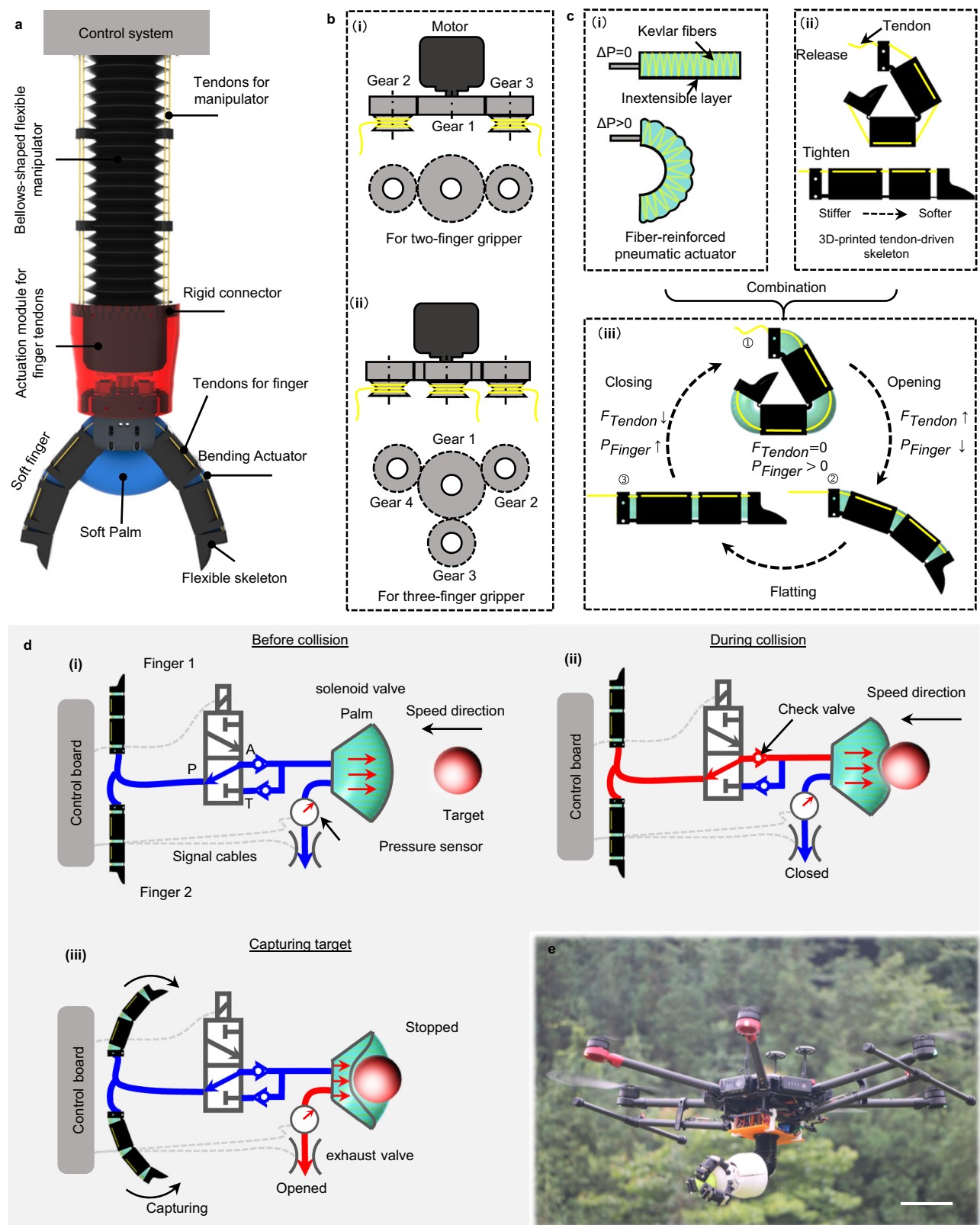

energy absorption and further inspire the optimal design to catch the incoming object with different velocities. This model can capture the dynamics of the palm/ball and gas exchange (Supplementary Fig. 2). Our simulation set-up is shown in Fig. 2a. The ball and palm are considered point masses, $m_{ball}$, and $m_{palm}$, and their collision is inelastic. The mass center of the palm is located at the top of the palm gas chamber, supported by internal damping force and static force resulting from the palm elastic force and internal gas pressure $P_{palm}$. The damping force is assumed to be linear to palm velocity. We correlated the static force with static experiment results, including complicated palm deformation and varying effective gas-pressurized areas (Supplementary Fig. 2, Supplementary Fig. 3). In practice, we parameterize different palm displacements and pressures and collect the force under each data point, further use these data to regress force as a

**Fig. 1 | Overview of the proposed soft gripper and the FPSED mechanism. a** A soft gripper consists of multiple soft fingers, an actuation module for finger tendons, a bellows-shaped flexible manipulator, and a control system. These fingers are actuated by tendons driven by a brushless DC motor in the actuation module. The fingers and palm are pre-inflated, and a valve system between the fingers and the palm controls the gas. A manipulator is used to adjust the grasping direction of the gripper. **b** An actuation module consists of a motor and several gears. Gear 1 drives the surrounding planetary gears (gears 2, 3, and 4) and the winding discs on them to rotate to realize the retraction and release of the tendons. (i) The two-finger gripper has one central gear (gear 1) and two planetary gears (gear 2 and 3), (ii) while the three-finger gripper only needs to add one planetary gear (gear 4). **c** A soft finger is mainly composed of (i) a fiber-reinforced pneumatic soft actuator and (ii) a 3D-printed tendon-driven skeleton. (iii) The fiber-reinforced actuator is inserted into the skeleton to build a soft finger whose grip and release are controlled by tendons and whose performance is enhanced by air pressure. **d** Three essential stages of the FPSED mechanism: (i) before collision, (ii) during collision, and (iii) capturing target. **e** A six-rotor drone integrated with the proposed soft gripper is presented and demonstrates its ability to capture dynamic targets outdoors. Scale bars represent 0.2 m.

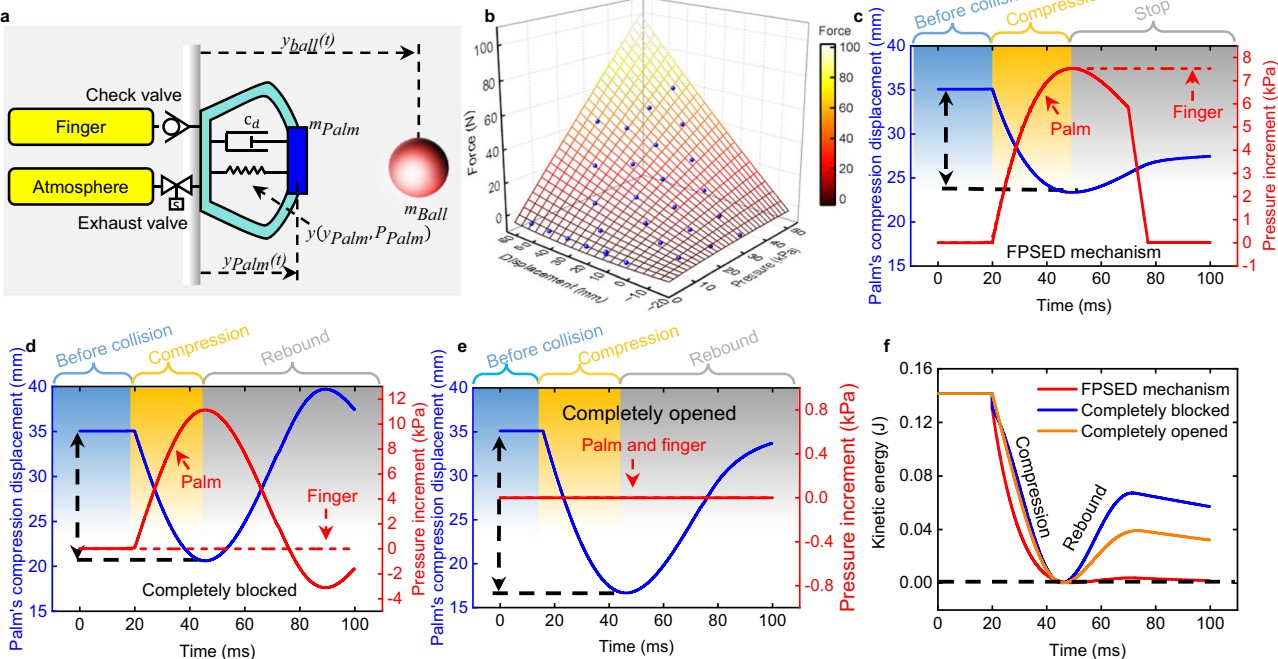

**Fig. 2 | Simulation set-up and results for the multi-physics model coupling palm/ball dynamics along with the gas exchange. a** Simulation set-up. **b** Fitted forces at different palm displacement and pressure data points. **c** FPSED mechanism. The check valve connects the palm and fingers for energy harvesting, and the controllable exhaust valve connects the atmosphere to dissipate the internal energy of the palm. **d** Completely blocked. The check and exhaust valves are completely blocked, and there is no gas exchange during a collision. **e** Completely opened. The check valve and exhaust valve are fully open to the atmosphere. We assume that the palm is not connected to any valve. **f** Comparison of predicted palm energy dissipation performance results for three valve states.

function of pressure and deformation. Such a mapping is shown in Fig. 2b. This function can be directly applicable to the dynamical simulation.

$$F_{\text{corre}} = H(P_{\text{palmair}}, x_{\text{palm}}) \tag{1}$$

where $F_{\text{corre}}$ is the correlated static palm elastic force regressed from data samples, $H$ is the mapping function, $x_{\text{palm}}$ is the palm displacement and $P_{\text{palmair}}$ is the palm's internal air pressure.

Regarding gas exchange, all gases are assumed to be an ideal gas. When the palm is compressed, the rising pressure pushes the gas towards the finger chamber through a check valve and connecting pipe. We model the finger-palm gas mass flow $\dot{m}_{\text{pipe}}$ as laminar viscous incompressible pipe flow in an equation since the pressure difference is slight and the connecting pipe provides the major resistance in the circuit.

$$\dot{m}_{\text{pipe}} = \pi(P_{\text{palm}} - P_{\text{finger}})\rho_{\text{air}} d_{\text{pipe}}^4 / (128 L_{\text{pipe}} \mu_{\text{air}}) \tag{2}$$

where the $P_{\text{finger}}$ is the finger air pressure, $\rho_{\text{air}}$ is the air density, $\mu_{\text{air}}$ is the air viscosity, $d_{\text{pipe}}$ and $L_{\text{pipe}}$ are pipe diameter and length.

The other solenoid valves connecting the palm and atmosphere are controllable. A high-pressure difference is expected when the valve

opens. Therefore, we use the adiabatic compressible flow rate equation for the flow rate calculation.

$$q_e = A_e p_0 \sqrt{\frac{2}{RT}\left(\frac{\gamma}{\gamma-1}\right)\left(\frac{p}{p_0}\right)^{\frac{\gamma}{\gamma-1}}\left[\left(\frac{p}{p_0}\right)^{\frac{\gamma-1}{\gamma}} - 1\right]} \tag{3}$$

where $\gamma$ is the air heat capacity ratio, $R$ is the gas constant, $T$ is the palm air temperature, $A_e$ is the cross-sectional area. $p$ is upstream pressure (palm air pressure in our case). $p_0$ is the downstream air pressure, which is the atmosphere.

Our simulations analyze the compression displacement and air pressure increment when the target hits the palm. Results reported in Fig. 2c, f show that, with the FPSED mechanism, we can harvest the impact energy to enhance the grip and completely dissipate the remaining energy to prevent fast rebounding. During compression, palm volume decreases with increasing compression displacement, rapidly increasing the air pressure within the palm and fingers. At the end of compression, by quickly opening and closing the exhaust valve, the internal energy in the palm is quickly released, and the palm's deformation will not be restored. Comparing the results in Fig. 2d, e, we found that this is the key to preventing objects from bouncing. Suppose the check valve and the exhaust valve are completely closed

(Fig. 2d) or always open to the atmosphere (Fig. 2e). In that case, most of the object's kinetic energy will be recovered from the potential energy of the palm. This result shows that it is almost impossible to completely dissipate the collision energy of an object only by squeezing the gas in the palm or the material of the palm. Correspondingly, our simulation results indicate that the FPSED mechanism exhibits an excellent potential for dissipating collision energy.

## Comparison of impact energy dissipation characteristics of several typical materials and palm designs

We design and experimentally compare four typical palms to illustrate that traditional grippers are unsuitable for capturing dynamic targets (Fig. 3b, Supplementary Fig. 6). The four designs have the same structural features and dimensions. We consider Design A a rigid palm made of 3D printed polylactic acid (PLA), like a traditional rigid gripper. Design B is fully solid silicone rubber with a shore hardness of 5 that acts like a regular shock-absorbing rubber pad. Designs C and D are made of the same materials as Design B but are structurally hollow airbags with a wall thickness of 2 mm. The difference between them is that design C is completely blocked in a test, and design D is fully open and directly connected to the atmosphere. A series of experiments is carried out on a self-built experimental platform to characterize these designs (Fig. 3a). We adopt a linear cylinder to launch a smooth glass ball to the palm along a smooth slideway. The smooth glass ball with a mass of 0.283 Kg and a diameter of 60 mm is used as the target for all collision tests. It should be clarified that our research aims to capture a target with kinetic energy by a new soft gripper. The target's kinetic energy is determined by its velocity and mass. Changing the velocity or mass of the object can obtain different kinetic energy. Here, we chose a relatively simple approach that alters the object's velocity to acquire various kinetic energy for quantitative comparison.

It is almost impossible to completely stop a high-speed target simply by the material properties of a palm itself (Supplementary Movie 1). For comparison, we define the total energy dissipation efficiency $\eta_t$ as the ratio of the dissipated kinetic energy $K_d$ of a ball during a collision to the initial total kinetic energy $K_t(\eta_t = \frac{K_d}{K_t} \times 100\%)$. We record the residual kinetic energy of a ball when it is separated from the palm after a collision as $K_r$, then $K_d$ can be obtained by $K_d = K_t - K_r$. Based on the measured kinetic energy data, the $\eta_t$ of Design A and Design D are relatively close after the collision, 83.3% (Supplementary Fig. 6a) and 82.5% (Supplementary Fig. 6d), respectively. This is superior to design B (50%) (Supplementary Fig. 6b) and design C (33%) (Supplementary Fig. 6c). The rigid palm (design A) converts kinetic energy primarily into internal energy dissipation. If the kinetic energy of the collision is high, the palm will attenuate the kinetic energy at the cost of damaging its structure. Collision energy is mainly converted into elastic potential energy for the soft elastic palm (designs B, C, and D). Once compressed to a maximum displacement, the stored elastic potential energy is quickly released to bounce the glass ball. Design D is also a soft elastic palm. However, its elastic potential energy accumulated is relatively small compared with the first three designs (Supplementary Fig. 6d) due to the ball rapidly squeezing the palm resulting in an exhaust during the collision. In the process, most of the energy dissipation by pressure release and shear friction between the gas and the inner wall of the exhaust pipe is irreversible. After compression to the maximum displacement, the ball starts to rebound. Part of the rebound energy is recovered from the gas returning to the palm from the atmosphere. Therefore, these results demonstrate that transferring internal air pressure outward when an object collides with a soft palm is highly beneficial for improving the energy-absorbing capacity of the palm.

## Performance of the FPSED mechanism

**Energy harvesting and dissipation capability.** The FPSED mechanism theoretically manifests a superior energy dissipation capacity to avoid

rebound. We also observed experimentally that the energy of a ball could be almost dissipated entirely, and the experimental results are shown in Fig. 3c–e and Supplementary Movie 2. In the FPSED mechanism, there are two designed energy dissipation pathways, one is to transfer part of the energy to the finger, and the other is to dissipate by releasing gas. In addition, the rest of the kinetic energy of the ball is dissipated by compressive deformation of the elastic palm, system vibration, friction, etc. We define the energy harvesting efficiency $\eta_h$ as the percentage of the harvested energy $E_h$ to the initial total kinetic energy $K_t$ of a ball ($\eta_h = \frac{E_h}{K_t} \times 100\%$). The dissipated energy $E_d$ by releasing gas as a percentage of the total energy is defined as the energy dissipation efficiency $\eta_d(\eta_d = \frac{E_d}{K_t} \times 100\%)$. Further, besides the above two factors, the residual kinetic energy dissipation of a ball also depends on the material and structure of a palm, as well as vibration and friction.

To measure the $E_h$ and $E_d$, we first collect a set of kinetic energy data based on the FPSED mechanism through collision experiments (Fig. 3c(i)–e(i)). Then, with the exhaust valve always closed, we obtain a set of kinetic energy data without energy harvesting (Supplementary Fig. 5). Finally, with both the exhaust and gas transfer pipes closed, we collect a set of kinetic energy data without energy harvesting and air release (Supplementary Fig. 6c). For these three sets of data, the remaining kinetic energy is extracted when the ball is separated from the palm, which is recorded as $K_{r1}$, $K_{r2}$, and $K_{r3}$. Because the experiential results of collisions with three different initial kinetic energies show that $K_{r1} = 0$, $E_h$ and $E_d$ can be obtained by $E_h = K_{r3} - K_{r2}$ and $E_d = K_{r2}$. Quantitatively, for the FPSED mechanism, the experimental results with three different initial kinetic energies (0.14 J (Fig. 3c(i)), 0.57 J (Fig. 3d(i)), and 0.88 J (Fig. 3e(i))) demonstrate that their energy harvesting efficiencies $\eta_h$ are 27.86, 29.82, and 31.82%. Their energy dissipation efficiencies $\eta_d$ are 41.43, 31.58, and 26.13%, respectively. Noticeably, these results perfectly agree with the simulation results computed by the model we built above. These results illustrate that the greater the kinetic energy of the incoming target, the greater the energy harvested by the FPSED mechanism. This reduces the energy that needs to be dissipated by deflation. Under the relatively large kinetic energy impact, we observed that the increased deformation of the palm increases the proportion of energy dissipated by the palm deformation. Importantly, benefiting from the FPSED mechanism, the initial total energy can be dissipated completely.

**Compatibility and scalability.** The compatibility of the FPSED mechanism affects whether a gripper can capture impactors with different kinetic energy under a certain system parameter. The allocation of energy harvesting and dissipation efficiency is adaptive to an object in this mechanism. Under the same experimental conditions, objects with different initial collision kinetic energy are considered for testing compatibility. As demonstrated in Fig. 3c–e and Supplementary Movie 2, after the glass ball hits the palm at the speed of about 1 and 2 m/s, respectively, their energy can be fully dissipated (Fig. 3c(i), d(i)), and they stop immediately after the impact (Fig. 3c(ii), d(ii)). However, as the collision velocity increases to 2.5 m/s, the ball rebounds at a relatively small velocity of about 0.16 m/s (Fig. 3d(i), d(ii)). In this case, it is logical that the ball should stop completely since the FPSED mechanism's energy dissipation efficiency is high (Fig. 3g) and there is little energy left over that has to be released (Supplementary Fig. 7). From the time-state results during the collision (Fig. 3e(ii) and Supplementary Fig. 3d), the top and side of the palm overlap formed by the excessive deformation of the palm (more than 40 mm) at the late stage of impact is the main reason for this phenomenon. These results indicate that a palm can be well-compatible with objects in a specific range of collision kinetic energy, but such compatibility is usually limited.

An efficient method presented in this work for scaling up the performance of the FPSED mechanism is to control the pre-charged

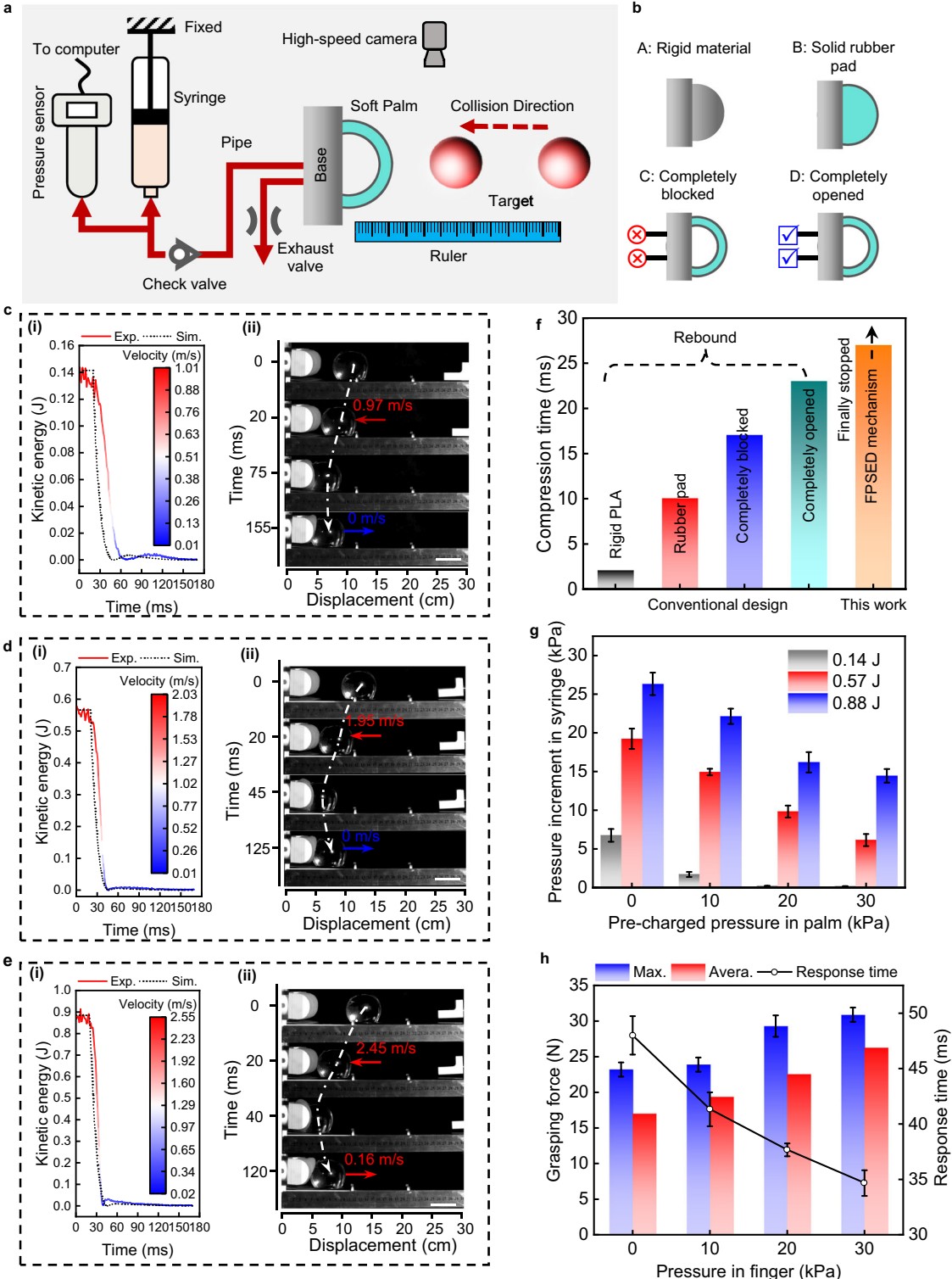

**Fig. 3 | Palm characterizations. a** Schematic diagram of the experimental set-up for collision tests. **b** Four typical palm designs. Design A is a rigid palm printed with polylactic acid; design B is a solid silicone rubber pad; designs C and D are both hollow soft airbags. The difference between them is that the vents of design C are completely closed, while design D is always open to the atmosphere.
**c**–**e** Experimental results of the proposed FPSED mechanism at three collision kinetic energies 0.14 J (**c**), 0.57 J (**d**), and 0.88 J (**e**). Left image (i) is the kinetic energy dissipation curve of the ball, where the color map represents the velocity decay of a ball. The picture on the right (ii) shows the position of the ball-palm at each key time point. **f** Comparison of the collision compression time between the conventional

four palms and the proposed FPSED mechanism. **g** Influence of initial collision kinetic energy of a target and pre-charged pressure in a palm on energy harvesting. We measure the pressure increment in a syringe after three kinds of collision kinetic energy under four kinds of pre-charged pressures. **h** Effect of positive pressure of bending actuator on finger grasping force and response time. The column shows the grasping force of a single finger on a 60 mm diameter cylinder. The dot-line graph represents the response time of the finger at different pre-inflation pressures. Scale bars in **c**, **d**, **e** represent 50 mm. Error bars represent the standard deviation of three measurements.

pressure in the palm. It enables the FPSED mechanism to balance energy harvesting and dissipation efficiency, thus ensuring superior compatibility with different kinetic energy objects. As shown in Supplementary Fig. 8a–i, a pre-charged pressure of 10 kPa can immediately stop the balls at speeds ranging from 1 to 2.5 m/s. The 2.5 m/s ball can be effectively stopped within the pre-charged pressure from 10 to 30 kPa (Supplementary Movie 3). Our experiments and simulations are well validated when considering the pre-charged pressure. Besides, the simulation follows a similar trend to the experiments for higher ball velocity with high pre-charged palm pressure (Supplementary Fig. 8h, i). We analyze the experimental videos and find that the palm deforms dramatically and rubs the ball during the rebounding stage, generating additional drag to slow it down. Although our simulation model could not capture the rubbing effect, it is reasonable to provide a theoretical point of view to explain the FPSED mechanism. These experimental and theoretical results confirm that the performance of the FPSED mechanism is scalable, and we can easily adjust the pre-inflated pressure in the palm to tune its energy harvesting and dissipation properties.

More importantly, in the FPSED mechanism, we expect more energy to be harvested rather than dissipated by releasing gas. If a target with relatively small kinetic energy is captured with a high-pressure palm, the energy harvesting efficiency will be reduced, which will cause more energy to be dissipated by releasing gas. Theoretically, based on the FPSED mechanism, this system can adaptively maintain the balance between energy harvesting and dissipation according to the target kinetic energy and the pressure control so that the objects of various speeds do not rebound after a collision. In practice, the commercial exhaust valve has a limited vent size and response time, with high positive pressure in the palm. If the collision speed is relatively low, the proportion of energy dissipated by releasing gas can be very high. We have to adopt an exhaust valve with a larger vent size and faster response speed. Experimental results of the ratio of the energy dissipated by releasing to the initial total energy under different pre-charged pressures and collision velocities are shown in Supplementary Fig. 7. For a low kinetic energy collision (0.14 J), if the palm is inflated with additional pressure (from 10 to 30 kPa), the energy that needs to be dissipated by releasing gas accounts for more than 60% of the total energy. In particular, the proportion at 30 kPa exceeds 80%.

**Ability to prolong collision duration.** The FPSED mechanism has outstanding performance in prolonging the collision duration, which provides sufficient response time for a gripper to capture dynamic targets. Taking the pre-inflation of the palm as 0 kPa and the ejection velocity of the ball as 2 m/s, by analyzing the FPSED mechanism and the four typical palms designed above (Fig. 3f), consistent with our expectation, soft materials have larger deformation displacements than rigid materials and therefore longer compression times. Notably, the compression time (from the ball touching the palm to the maximum compression displacement) under the FPSED mechanism is 12 times that of a rigid palm. Although this compression time is only tens of milliseconds more prolonged than the compression time corresponding to designs b, c, and d, there is an essential difference among them. The ball under the FPSED mechanism always keeps in contact with the palm after the collision. Nevertheless, with the other three designs, the ball bounced within milliseconds. We believe that the performance of the FPSED mechanism to extend the collision duration is superior, which is difficult to achieve in other designs.

**Effect of harvested energy on fingers.** After solving how to stop a dynamic target quickly and maintain a longer contact time, the gripper possesses the key and primary conditions for grasping a dynamic target. Next, we evaluate how the energy harvested during a collision improves a finger's grasping force and response time.

By gathering the data from the pressure sensor used in the collision experiment described above, we can first determine how much the air pressure in a closed cavity is raised during a collision. We then plot three initial kinetic energies of 0.14, 0.57, and 0.88 J against the pressure increment in the syringe at various pre-inflation pressures (Fig. 3g). These findings suggest that more gas will be delivered when the target's collision kinetic energy is higher and the precharge pressure is constant. When the pre-charged pressure increases, the gas transfer capacity decreases. For example, when the pre-charged pressure is 0 kPa, the pressure increment corresponding to the collision kinetic energy of 0.14 J is close to 7 kPa. When the kinetic energy increases to 0.88 J, the pressure increment is significantly increased to 26 kPa. In contrast, when the pre-charged pressure is 30 kPa, the pressure increment drops to 15 kPa.

Then, based on the above results, we change input air pressure from 0 to 30 kPa with increments of 10 kPa into the bending actuator to evaluate the effect of the air pressure transmitted by the collision on the grasping force. We pull a 60-mm-diameter cylinder of negligible mass from the fully its fully wrapped state. The variation of this pulling force can characterize the grasping ability of fingers (Supplementary Fig. 9a). These results (Supplementary Fig. 9b–e) reveal a general pattern for how objects separate from fingers: steady grasping, slipping, and sudden detachment before it can completely escape from the finger. Referring to our previous research[33], we certified that the stiffness gradient on the finger is a critical reason for the evolution of the grasping force that first increases and then slowly decreases and finally disengages. The end and middle joints are essential in resisting object displacement in the steady grasping stage. However, with the further displacement increase, the object quickly slipped from the high-stiffness end joint and middle joint to the low-stiffness middle and fingertip joint. Especially in the early stage of complete detachment, only the fingertip is left to function. In the steady grasping stage, the grasping force gradually increases while the target object moves and the target surface is in close contact with the finger surface. After starting to slip, the grasping force gradually decreases until the maximum detachment force is reached, during which the target rolls on the inner surface of the finger. As shown in Fig. 3g, without additional inflation, the object transitions from a steady to a non-steady state starting at 23.2 N and disengages completely from the fingertip when the force drops to 14.1 N. In contrast, when the pressure in the bending actuator rises to 30 kPa, the object starts to slip at 30.9 N, and a more significant force of 26.6 N is required to drag the object to disengage it. The average forces in the slip area at the air pressure of 0, 10, 20, and 30 kPa are 17.0, 19.5, 22.6, and 26.3 N, respectively.

Eventually, we characterize that the harvested energy accelerated the finger's response time while improving the grasping force. To capture fast dynamic objects, it is critical that the gripper has the ability to overcome the "fast" of dynamic objects. As shown in Fig. 3f and Supplementary Fig. 6a–d, the collision between the ball and the palm from contact to complete disengagement is relatively short, which requires the gripper to have a fast response time to capture. In fact, it is challenging for the reported soft gripper to meet such high responsiveness requirements. Here, we employ a high-speed camera (1000 fps) to measure the time it takes for a finger to be released from a fully flattened state until the finger wraps around a 60 mm diameter cylinder. Experimental results in Fig. 3h show that with no additional air pressure in the fingers, it took 48 ms for the system to transit from fully flattening to grabbing a 60 mm diameter cylinder. As the pressure inside the finger increases, the time corresponding to the same behavior is significantly shortened. For example, when the pressure is 30 kPa, the response time is improved by 28%, and it only takes 34 ms to capture the object.

Taken together, objects with more kinetic energy tend to require greater grasping force and response time from the fingers. In this

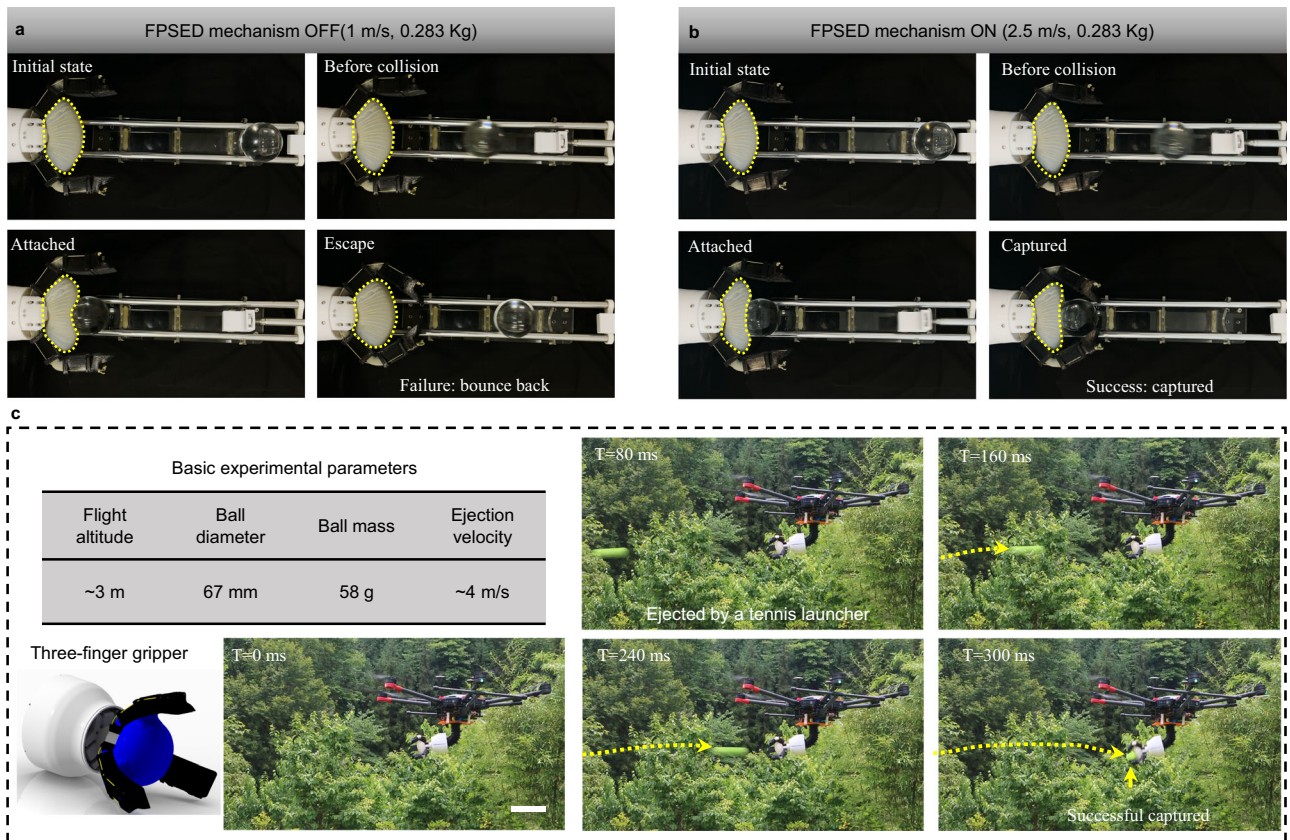

**Fig. 4 | Soft gripper capture demonstrations. a** With the FPSED mechanism turned off, the two-finger gripper has difficulty capturing relatively low-speed (1 m/s) dynamic targets. **b** After the FPSED mechanism is turned on, the two-finger gripper can successfully capture a relatively high-speed (2.5 m/s) dynamic target. In both tests, the ball is placed on a horizontal rail with a mass of 0.283 Kg. **c** Demonstration of a six-rotor drone equipped with a three-finger soft gripper to capture dynamic targets in an outdoor environment. A launcher ejects the tennis ball as a target, and it has a mass of 58 g, a diameter of 67 mm, and a speed of ~4 m/s. Scale bars represent 0.2 m.

study, benefiting from the FPSED mechanism, the finger grasping force and response time are positively related to the kinetic energy of the objects, which significantly reduces the difficulty of control and improves grasping stability.

## Potential applications

We fabricate a two-finger gripper prototype to demonstrate that the FPSED mechanism can empower a soft gripper to capture dynamic targets (Fig. 1a). In these demonstrations, the target ball has a diameter of 60 mm and a mass of 0.283 Kg. We let the gripper capture objects with different launch velocities, and their initial kinetic energies are set to 0.14, 0.57, and 0.88 J, respectively. With the palm and end joint glued together, we pre-inflated the palm in each set of experiments to prevent a partial collapse of the palm when the fingers are opened. Here, three experiments are pre-charged with 5, 10, and 15 kPa from low kinetic to high kinetic energy, respectively. Based on the above argument, we let the sizeable kinetic energy collision correspond to a higher pre-charged pressure to improve the compatibility of the gripper. Given the 1 m/s collision experiment as an example, once the FPSED mechanism is not turned on (Fig. 4a), even if the ball hits the gripper with a relatively low kinetic energy, the gripper is utterly incapable of capturing it, let alone a higher speed collision. However, once the FPSED mechanism is turned on (Fig. 4b), the gripper effortlessly captured a dynamic target with a kinetic energy of 0.88 J (2.5 ± 0.05 m/s) (Supplementary Movie 4), even though it is only 120 ms from the target launch to hit the palm. Similarly, we demonstrate the ability of the gripper to capture dynamic targets with other kinetic energies.

To highlight the capabilities of our FPSED mechanism in practical applications, we integrate a three-finger gripper with FPSED mechanism into a hexacopter drone and conduct experiments outdoors for demonstration (Fig. 4c). Capturing with a drone is very challenging, mainly because it requires the drone to achieve the dual goals of manipulation and maintaining a stable flight. In our study, we first use a remote control to take off the drone to a height of three meters and keep it hovering. Then, we adjust the bending angle of the flexible manipulator to an appropriate position (horizontal forward). Next, we launch a 58 g weight tennis ball with a 67 mm diameter to the soft gripper by a tennis transmitter. As expected, after the target hits the palm, the FPSED mechanism is triggered, and the gripper immediately captures it. By analyzing the recorded video (Fig. 4c, Supplementary Movie 5), the tennis ball is firmly captured by the gripper without rebound after hitting the palm, and the flying attitude of the drone is not affected. This proves that the soft gripper is feasible for capturing dynamic targets, and the FPSED mechanism can ensure safe, stable, and reliable capture.

## Discussion

To summarize, these capture experiments demonstrate the use of the FPSED mechanism to harvest and dissipate collision energy with sufficient energy dissipation efficiency to achieve deceleration and with sufficient collision time for captures. These grippers with the FPSED mechanism have wider compatibility for collision speed during capturing. A highlight of the FPSED mechanism is that its capabilities are adjustable and extendable simply by setting the pre-inflated pressure. This mechanism can be designed in various configurations to adapt to

rigid machines, soft equipment, etc. These properties will be important for robotic grasping in some environments where collisions are difficult to avoid.

To improve the capturing performance of soft robotic grippers on dynamic targets, future research directions include improving the intelligence of the soft gripper and reducing the redundant components of the FPSED mechanism. System intelligence/perception capabilities are also crucial because it is necessary to predict kinetic energy, pose, and trajectory before capturing a dynamic target. The intelligence improvement can be investigated through machine learning and computer vision, enabling the intelligent collaboration of fingers, palms, targets, and carriers. In order to reduce the redundancy of the FPSED mechanism, future research can combine soft valves[34,35], pumps[36,37], and flexible sensors[38,39] to realize that some rigid devices in the FPSED mechanism are entirely replaced and integrated into a soft device. From the perspective of robot design and application, expanding or reducing the palm size can derive more applications but also bring related challenges. The larger size introduces more deformation and gas volume, allowing higher-speed targets to be capturable by the palm. The smaller size makes it possible for the FPSED mechanism to be extendable to some micro-robots and aircraft. In particular, the minor size requirement presents manufacturing challenges. More broadly, our work shows that soft grippers can fully dissipate the energy of dynamic targets and capture them efficiently and stably. These properties are important for developing next-generation soft robots for various challenging applications such as space debris cleanup, failed spacecraft recovery, aerial manipulators, etc.

## Methods

### Fabrication of the proposed soft gripper

The palm and bending actuators are manufactured similarly but have different structural designs (Supplementary Fig. 1a–c). For a palm, the saddle-like structure makes it easier to deform during a collision because we expect the kinetic energy dissipation efficiency of the target object to be positively correlated with the deformation of the palm. However, the excessive expansion and deformation of the palm can easily lead to the rapid accumulation of energy. In the rebound stage of the collision, the accumulated energy will react to the target object, prompting the object to rebound quickly. Therefore, we adopt high-strength Kevlar fibers to constrain the palm. The neutral layer of the palm wall contains these fibers that intertwine, allowing the saddle-shaped palm to expand along the finger opening and closing plane but not along the vertical plane. Like fingers, we inflate the chamber of the palm to maintain its appropriate initial stiffness for matching the palm's stiffness and the target's kinetic energy, thereby expanding the gripper's adaptability to various objects' kinetic energy and the application scenarios.

The manufacturing process of the palm and bending actuator is the same, and both are cast in the corresponding 3D printed mold (Supplementary Fig. 1). It mainly includes four steps. The first step is to cast the inner layer structure, and the surface of this layer forms a groove structure for winding. The second step is to cross-wrap the Kevlar wire at an angle of 9°. At this time, a layer of soft but the non-stretchable non-woven fabric needs to be placed on the bottom surface of the bending actuator to limit the axial extension of the actuator. The third step is to pour the outermost layer. Finally, take out the central shaft and seal the two end faces. After each pouring, the sample should be placed in a 60 °C oven for 2 h for the liquid silicone rubber to cure.

Another finger component is the flexible skeleton, made from 3D-printed thermoplastic polyurethane (TPU) rubber (Supplementary Fig. 1d). The hardness of TPU is 95 A, and the elongation at break is ≥800%. The detailed printer parameters are set as in our previous research[33]. The joint thickness adjusts the stiffness of each joint of the skeleton. The smaller the thickness, the smaller the stiffness. The thickness of this study's distal joint, middle joint, and fingertip joint are 5, 3, and 1 mm. Finally, we inserted the bending actuator into the flexible skeleton and glued the bending actuator's end face to the skeleton's end face together.

### Valve system

The valve system includes a 3/2-way solenoid valve (SNS, 3V210-08), two check valves (AQUA TECH, AQTCV), an exhaust valve (2W030-08, AirTAC), a pressure sensor (XGZP6847A, CFSensor Ltd.), and a control board (Arduino 2560). One of the two exhaust pipes at the bottom of the palm is connected to the exhaust valve through a pressure sensor, the other is connected to the air inlet A of the solenoid valve, and the air outlet P is connected to the air pipes of the two fingers. When the fingers are closed, the gas can only flow from the palm to the fingers, and when the fingers are opened, the gas flows in the opposite direction. This process is realized through the two one-way valves. The Arduino microcontroller reads the value of the pressure sensor and sends commands to the metal-oxide-semiconductor field-effect transistor of the solenoid valve and exhaust valve, respectively.

### Dynamics model

We build a multi-physics model to predict the palm/ball dynamics and gas exchange. As shown in Supplementary Fig. 2, we separate the entire ball-approaching and vacating process into 3 stages: (1) Before collision; (2) Attached, ball and palm motions are synchronized before the ball detaches; (3) After detached, ball and palm move independently. For stage 1 and stage 3, the palm and ball are moving separately. The ball only has friction force with the rail, and palm undertakes the damping force and the correlation static force $\mathbf{F_{corre}}$ (introduced in Fig. 2b). Differential dynamics equations for the palm and ball motion are as follows:

$$\begin{bmatrix} m_{\text{ball}}\ddot{\mathbf{x}}_{\textbf{ball}} \\ m_{\text{palm}}\ddot{\mathbf{x}}_{\textbf{palm}} \end{bmatrix} = \begin{bmatrix} -\mu m_{\text{ball}}\mathbf{g} \\ -D\dot{\mathbf{x}}_{\textbf{palm}} + \mathbf{F_{corre}}\left(P_{\text{palm}}, x_{\text{palm}}\right) \end{bmatrix} \quad (4)$$

where $\mu$ is the friction coefficient between rail and ball and $D$ is the damping coefficient of the palm.

Once the two objects come into contact, they are attached and considered stage 2. For stage 2, two objectives are moving together. The differential dynamics equations for the palm and ball motion are as follows:

$$(m_{\text{ball}} + m_{\text{palm}})\ddot{\mathbf{x}}_{\textbf{together}} = -\mu m_{\text{ball}}\mathbf{g} - D\dot{\mathbf{x}}_{\textbf{palm}} + \mathbf{F_{corre}}(P_{\text{palm}}, x_{\text{palm}}) \quad (5)$$

where $\ddot{\mathbf{x}}_{\textbf{together}}$ is the acceleration of both palm and ball when they are moving together.

Once the palm decelerates in the rebounding direction, meaning the palm cannot push the ball anymore, the objects detach, and the process enters stage 3, when the dynamics Eq. (4) can be reused.

Meanwhile, we calculate the gas flow and pressure change in the system. The equation for finger-palm gas flow:

$$\dot{m}_{\text{pipe}} = \pi(P_{\text{palm}} - P_{\text{finger}})\rho_{\text{air}}d_{\text{pipe}}^4/(128L_{\text{pipe}}\mu_{\text{air}}) \quad (6)$$

The equation for solenoid valve gas flow:

$$q_e = p_0\sqrt{\frac{2}{RT_i}\left(\frac{\gamma}{\gamma-1}\right)\left(\frac{p_i}{p_0}\right)^{\frac{\gamma}{\gamma-1}}\left[\left(\frac{p}{p_0}\right)^{\frac{\gamma-1}{\gamma}} - 1\right]} \quad (7)$$

where the variables are defined in the main manuscript.

The volume of a finger is assumed to fix, and the volume of the palm is assumed to be an affine function towards the palm length (in

practice, we use palm mass center displacement as we consider the palm base as the origin), such that:

$$V_{\text{palm}} = \frac{V_{\text{palm0}} x_{\text{palm}}}{x_{\text{palm0}}} \tag{8}$$

Here $V_{\text{palm0}}$ and $x_{\text{palm0}}$ are the palm volume and displacement at the initial condition.

Considering the gas within the palm as an open system with varying volumes, we have

$$\frac{dP_{\text{palm}}}{dt} = -\frac{\gamma_{\text{air}}{}^{*}m_{\text{palmair}}}{V_{\text{palm}}{}^{*}P_{\text{palm}}} * \left( m_{\text{palmair}} \frac{dV_{\text{palm}}}{dt} + V_{\text{palm}} * \frac{\left( -\frac{dm_{\text{pipe}}}{dt} + \frac{dm_{\text{valve}}}{dt} \right)}{m_{\text{palmair}}^{2}} \right) \tag{9}$$

Similar to the finger gas open system with fixed volume:

$$\frac{dP_{\text{finger}}}{dt} = -\gamma_{\text{air}}{}^{*}P_{\text{finger}} * \left( m_{\text{fingerair}} \frac{dm_{\text{valve}}}{dt} \right) \tag{10}$$

where $\gamma_{\text{air}}$ is the heat capacity ratio. The subscript denotes the other mass, pressure, and volume terms.

Overall, our differential equations have eight unknowns: the velocity, acceleration of the palm and ball, and the pressure and mass of the air within the palm and finger. The equation is solved by forward Euler.

## Experimental set-ups

**Quasi-static experiments.** In order to predict the palm force under collision and gas pressure in the simulation model, we design a quasi-static experiment to measure the relationship between palm displacement and pressure. A push-pull force transducer is mounted (DS2-100, ZHIQU Instrument Co., Ltd.) on a vertically placed linear guide. An indenter with a radius of 30 mm is fixed on the force transducer probe (Supplementary Figs. 3a and 4a). The palm is mounted on a base, and the exhaust tube of the palm (inner diameter 6 mm, length 200 mm) is connected to a syringe with a capacity of 60 ml. A one-way valve between the palm and the syringe ensures that airflow cannot flow from the syringe to the palm, and a pressure sensor (SUP-PX300, Supmea Automation Co., Ltd) is used to measure the pressure in the syringe. During the experiment, the syringe's plunger is fixed at 30 ml. The moving speed of the guide rail is 10 mm/s.

First, we calibrate the initial position of the indenter under different pressures. The palm pre-inflation pressure increases from 0 to 30 kPa in 5 kPa increments. When the indenter moves down until the value of the force sensor is 0.1 N, we think that the indenter is just in contact with the palm. The position of the guide rail at this time is regarded as the initial position under the corresponding air pressure (Supplementary Fig. 3b).

Next, we calibrate the equivalent area of the indenter's contact with the palm (Supplementary Fig. 4b). In this experiment, the indenter is moved to the specified displacement, and the force sensor value $F_1$ and the pressure sensor value $P_1$ are recorded. Afterward, record the value $F_2$ of the force sensor once again while maintaining the indenter's location and connecting the palm's exhaust pipe to the atmosphere ($P_1$ becomes $P_2 = 0$ kPa). Using the difference between the force and the pressure in the process of the palm wall after decompression, we can calculate the equivalent area of the indenter and the palm. Four groups of pre-inflated pressures are set in this experiment: 0, 10, 20, and 30 kPa. For each set of pre-charged pressures, we measure the indenter's force and pressure changes (Supplementary Fig. 4c) at six displacements of 0, 10, 20, 30, 40, and 50 mm.

Finally, we test the effect on palm stiffness at different pre-charged pressures (Supplementary Fig. 3c, d). The indenter presses the palm down at a constant speed of 10 mm/s, and the force sensor record the force value during the displacement of the indenter from 0 mm to 50 mm. The pressure sensor records the real-time pressure inside the syringe. In this experiment, four groups of 0, 10, 20, and 30 kPa are tested. The force-displacement relationship for a set of palms with no pre-inflated pressure and the exhaust pipe fully vented to the atmosphere is measured to differentiate the effects of pressure and palm wall on palm stiffness.

**Collision experiments.** These experiments utilize an air cylinder (SC32×100, AIRTAC) to provide initial velocity to glass spheres (60 mm in diameter, 0.283 Kg in mass). The glass ball is placed on a smooth horizontal track, the center of which is 270 mm from the palm surface. The palm and its base are mounted on a lab bench. We implement three collision velocities to verify and compare five design cases. In all collision experiments, the speed and collision state of the ball is recorded by a high-speed camera (AcutEye-1M3-1000, Rock-eTech Technology Corp., Ltd.). Its frame rate is set to 1000, and the resolution is set to 1280 × 1024. When the air pressure of the cylinder is 60 kPa, the speed of the glass ball is $1 \pm 0.05$ m/s, 100 kPa corresponds to $2 \pm 0.05$ m/s, and 150 kPa corresponds to $2.5 \pm 0.05$ m/s. A video analysis software (Tracker, open-source physics) is used to extract the kinetic energy data of the ball from the collision video. The data presented are the mean of three experimental results.

**Grasping force.** To measure the improved effect of harvested energy on the gripping force, we experimentally measured the grasping force of a single finger. The design parameters of the finger skeleton and the bending actuator are shown in Supplementary Fig. 1, and the thickness of each joint from the finger base to the fingertip is 5, 3, and 1 mm, respectively. The experimental set-up is shown in Supplementary Fig. 9a. A 3D-printed cylinder with a diameter of 60 mm is used as the target, which is moved upwards under the drag of linear guides. A push-pull force sensor records the force evolution data in real time. Further, we plot the grasping force curves corresponding to the four pre-charged pressures (0, 10, 20, and 30 kPa), respectively, and mark the maximum grasping force, the maximum detachment force, and the average force in the slip stage (Supplementary Fig. 9b–e).

**Potential applications.** We demonstrate a two-fingered and a three-fingered gripper to capture dynamic targets. First, a two-finger gripper is designed, and the two fingers of the gripper are stacked in an arrangement with the palm between the two fingers, which are integrated by a 3D printed base. The tendon (Kevlar fibers) used to actuate the two fingers is driven by a motor (ROBOMASTER GM6020, DJI), and a pinion gear ($m = 0.8$, $z = 21$) is installed on the output shaft of the motor as a sun gear. Two planetary gear mesh ($m = 0.8$, $z = 30$) with it are symmetrically arranged, and a pulley is installed on each planetary gear for winding the tendon. A smooth glass ball is used as a target to shoot at the gripper at different speeds, and the capture process is recorded with a camera. Then, on this basis, a three-finger gripper is designed by us and mounted on a bellows-like flexible robotic arm to demonstrate the gripper's ability to grasp objects outdoors. Because the flexible manipulator is hollow, the power and signal cables and air pipes pass through the middle of the manipulator. The flexible manipulator's valve system, motor, and battery are installed in a control box. The gripper is installed on a six-rotor drone (M600Pro, DJI).

## Data availability
All the data supporting the finding of this study are available within the main text of this article and its Supplementary Information files or

from the corresponding authors upon request. Source data are provided with this paper.

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

## Acknowledgements

This work was supported by the National Natural Science Foundation of China, Project No. 52275206.

## Author contributions

W.P. and Y.Z. conceived the idea of the study. Y.Z. designed and fabricated the experimental platform and prototypes and demonstrated their functions. W.Z. contributed to modeling and performed the simulations. Y.Z. and P.G. conducted the experimental work. Y.Z. wrote the manuscript. All authors discussed the results and contributed to the final manuscript. W.P. and X.Z. supervised the study.

## Competing interests

The authors declare no competing interests.
