## [Peer Review File · Nature Communications]

REVIEWER COMMENTS

Reviewer #1 (Remarks to the Author):

The paper presents an interesting and original work on dynamic object gripping. The idea of dissipating the impact energy and harvesting part of it for improving the gripping action is original and valuable. The work notably shows the effectiveness of this idea.

The work is significant in the field of soft robotics, as the implementation of the idea is based on the use of soft materials and some soft robotics technologies. However, the significance of the work goes beyond it. The work is interesting for dynamic catching in general.

The work is presented thoroughly, from the idea to the theoretical analysis and to the experimental validation. The results are described in detail and they well support the starting idea. The experiments are focused on that. The overall methodology is good and make the results convincing.

The discussion is correct and it well elaborates on the specific case addressed here, with respect to the generality of the idea.

The paper is well-written and no improvements to the text are deemed necessary.

Reviewer #2 (Remarks to the Author):

This manuscript describes the development of a "soft" gripper designed for capturing objects with high kinetic energy. The mechanism is largely divided into the palm and fingers. The key component that makes the proposed gripper different (and better) at catching objects is the ability to dampen the impact from the pressure regulation. In addition, the excess pressure is used (harvested) into the fingers for improved response time.

Overall, I find the main idea interesting and the palm-finger combination seems to work well at capturing "glass balls". The modeling and experiments are convincing, except for the fact that it is mostly limited to glass balls. However, there are a few major issues to be clarified.

1. I find the descriptions of the palm, valves, and related components relatively clear. This covers the tests for absorbing the KE from the impact. What is really unclear is how the fingers work. This is not even briefed in the introduction. How exactly are they actuated? How many actuators are used? How are they controlled, manually by a user? At the beginning, because of Fig. 2, I thought the

fingers were passive as they did not move in Fig 2c,d,e. However, after reading further, there are tendons and muscles. I am rather lost here.

2. To expand from the question above, based on Fig. 1a, there is "one" motor, but I don't think that is used to control the tendons? What exactly are in the control system? In Extended Data Fig. 1, what are the hardware components for muscle A and muscle B?

3. I am asking because, from my understanding, the palm and fingers are the two main components. But the fingers are not adequately described (I know that some may have been published and cited as previous works, but at the moment, the manuscript does not make sense in itself). And without this information, I am not yet convinced that the improved reaction time is critical in catching the objects.

4. Following above, a palm designed with Design C (completely opened) seems to do quite well for dampening out the energy. With the leftover energy of less than 0.1 J, the glass ball bouncing speed is around 0.8 m/s(?), it would travel around 3 cm in 40 ms, would that be sufficient time for normal fingers to catch the ball? I cannot answer because I do not understand how the fingers are activated.

5. Another issue is the focus. Based on the introduction, the focus of the work is on damping out the KE for grasping. Suddenly, from Lines ~250, the consideration on releasing (ungrasp) and staility of a grasp is brought up without any context. I also do not see a tight connection with the main contribution of the work. I find it confusing and even distracting, let alone the fact that is difficult to follow or deduce the message the authors try to convey here.

===== Other Major points =====

- The use of the "magnet" analogy is not scientific nor useful. The gripper does not ACT like a magnet as it does not attract the object. Energy dissipation and harvesting are also not properties of magnetic forces.

- In the submitted version, there is no evident separation between the abstract and the main text. There are no section titles (Introduction/Results/Discussion). The separation is also unclear in the content of the main text. There also exists no headings for subsections in the Result "section". This makes it rather difficult to follow the manuscript. In fact, I tried to figure out where the transition from "Introduction" to "Results" is, but I cannot really pinpoint where.

- Line 192: "can reach 100%". I will be a little more careful with the claim as it sounds like it is always 100%. You might want to add "can reach 100% when ...", or make the claim less bold as "could/may reach up to 100%".

- Lines 196-200: How are "harvesting efficiencies" and "elastic potential" energy defined and measured?

- Pre-loading the palm with positive pressure allows an object with higher KE to be stopped completely. Is there a notable disadvantage of using high positive pressure in all cases (apart from lowering the harvesting efficiency)? Regardless of the answer, can you provide supporting data/measurements? (I guess this is Extended Data Fig. 6, but it would be great to summarize this point in the main text)
- I do not entirely agree with this sentence "We can assume that the collision duration of the FPSED mechanism is infinite" (Line 234). The collision time is until the object comes to a complete stop.
- The paragraph from Line 248 to 278 should be re-written. It starts with ".. to evaluate the effect of the air pressure transmitted by the collision on the blocking force and response time." But after talking about the blocking force, it transitions to "slipping and stable grasping" instead of talking about the response time (which eventually comes later). The content on stable grasping is also a little out of context as mentioned earlier.
- Line 264: "As shown in Fig. 3g, ..." I do not see what is described here in Fig. 3g.
- In the Conclusion, starting from line 332, I agree with "A highlight of the FPSED mechanism is that its capabilities are adjustable and extendable simply by setting the pre-inflating pressure,...". However, other points, such as "protecting targets and themselves", and "These properties will be important for robotic grasping in highly cluttered environments where collisions are difficult to avoid." are not clear and not demonstrated in this work. I do not believe you can make these conclusions.
- Line 336 "we added a 3D printed tendon-driven bellows-like flexible manipulator to the gripper." Is this bellow (actuated) structure solely for positioning/re-orienting the gripper? Please make it clear that it is an entirely separate component.
- Line 417-418 refers to air inlet A and air output P, but which figure are you referring to?
- Dynamics model: lots of variables undefined. P is pressure? How does one calculate f_{corre} ?

==== Minor (and editorial) comments =====

Abstract:

- "The robot can hardly absorb the impact energy" => perhaps it should be "a robot" as you talk about any robots.
- "almost 100%" => almost all
- "... impact energy in milliseconds (less than 30 ms)". => "impact energy in less than 30 ms".
- "...a three-finger gripper with a flexible 25 manipulator on a six-rotor drone flight platform to demonstrate..." => remove "flight platform".

Main text:

- Line 41: keeping their mechanical stability after perturbations.
- Line 78: remove "As we all know".
- Line 89: "...the rest gas will remain in..." => "the remaining gas would stay".
- Line 89: "The stored energy will be released rapidly when reaching the compression limit, which will cause the target to bounce back and escape quickly..." =>
- Line 136: several variables undefined in the main text (such as m_a).
- Line 156: what does "user-controllable" exhaust valve mean? What does a user control?
- The phrase "pre-charge" when used as an adj sometimes appears as "pre-charge", but sometimes as "pre-charged".
- Line 227: replace "common sense" with "expectation".
- Line 231: "Although the FPSED mechanism is only tens of milliseconds more prolonged than the designs b, c and d,..." This sentence must be re-written. The mechanism is NOT prolonged. The compression time is.
- Line 233: What is "limited position"? Do you mean "displacement"?
- Line 236: "dwell time" > "contact time".
- Line 247: Remove "when the collision kinetic energy is 0.88 J". It is redundant and not making the sentence any clearer.
- Line 313-230, approximately: the use of past/present tenses is not consistent.
- Fig. 4b, FPSED ON (1 m/s), the speed here does not match the text (2.5 m/s) (Line 307-308).
- Line 343: What is target's quality?
- Line 439, s.t. => such that
- Line 460-61, Kpa => kPa.
- Line 461: "When the indenter moves down until the value of the pressure sensor is 0.1 N", do you mean force sensor?

Responses to the reviewers' comments (NCOMMS-22-26383-T)

Reviewer #1 (Remarks to the Author):

The paper presents an interesting and original work on dynamic object gripping. The idea of dissipating the impact energy and harvesting part of it for improving the gripping action is original and valuable. The work notably shows the effectiveness of this idea.

The work is significant in the field of soft robotics, as the implementation of the idea is based on the use of soft materials and some soft robotics technologies. However, the significance of the work goes beyond it. The work is interesting for dynamic catching in general.

The work is presented thoroughly, from the idea to the theoretical analysis and to the experimental validation. The results are described in detail and they well support the starting idea. The experiments are focused on that. The overall methodology is good and make the results convincing.

The discussion is correct and it well elaborates on the specific case addressed here, with respect to the generality of the idea.

The paper is well-written and no improvements to the text are deemed necessary.

Response: We do appreciate your recognition of the originality of our research and positive comments on the manuscript. We hope the work will bring new perspectives of dynamical catching to other researchers in this field and potentially impact other communities as well. We are happy to hear your appreciation of our work and we will make persistent efforts in the future to further explore this direction.

Reviewer #2 (Remarks to the Author):

This manuscript describes the development of a "soft" gripper designed for capturing objects with high kinetic energy. The mechanism is largely divided into the palm and fingers. The key component that makes the proposed gripper different (and better) at catching objects is the ability to dampen the impact from the pressure regulation. In addition, the excess pressure is used (harvested) into the fingers for improved response time.

Overall, I find the main idea interesting and the palm-finger combination seems to work well at capturing "glass balls". The modeling and experiments are convincing, except for the fact that it is mostly limited to glass balls. However, there are a few major issues to be clarified.

Response: We do appreciate the reviewer's positive comments about our work and manuscript. We have addressed the comments, and detailed point-by-point responses to all comments can be found below.

1. I find the descriptions of the palm, valves, and related components relatively clear. This covers the tests for absorbing the KE from the impact. What is really unclear is how the fingers work. This is not even briefed in the introduction. How exactly are they actuated? How many actuators are used? How are they controlled, manually by a user? At the beginning, because of Fig. 2, I thought the fingers were passive as they did not move in Fig 2c,d,e. However, after reading further, there are tendons and muscles. I am rather lost here.

Response: Thank you for raising the question for clarification. We have added a more detailed description of the finger in the revised **Results** section "**Components and working principle of the gripper**", and also modified Fig. 1, shown below:

“Components and working principle of the gripper

The soft gripper developed in this study consists of multiple fingers and a soft palm, shown schematically in Fig. 1a. An actuation module is used to control the opening and closing of the fingers. In the actuator module, a motor and a planetary gear mechanism are applied to drive the tightening (fingers open) and releasing (fingers closed) of all finger tendons (Fig. 1b). We employ a motor-driven sun gear that meshes with the surrounding planetary gears for tendon tightening and releasing. For the

two-finger and three-finger grippers used in the following study, the difference in their actuation modules is the number of planetary gears (Fig. 1b(i) and Fig. 1b(ii)). To facilitate wirings, including pipes, signals, and power cables, a hollow, brushless direct-current (DC) motor with an integrated driver is adopted. Thus, all cables and pipes on the soft gripper can be routed through the motor shaft to the control system. Furthermore, to empower the gripper to capture in any direction, we design a 3D printed tendon-driven bellows-like flexible manipulator and connect it to the soft gripper with a rigid connector. Three tendons on the manipulator are arranged 120° along its circumference, and three servo motors are used to control the retraction and release of the three tendons in a winding manner, respectively.

Each finger has two main components. One is a soft fiber-reinforced bending actuator (Fig. 1c(i)), which is made of silicone rubber with a shore hardness of 5 degrees by casting and restricted by Kevlar fibers and an inextensible layer (Supplementary Fig. 1a). The Kevlar fiber constrains the expansion of the actuator along its radial direction, and the inextensible layer prevents it from extending axially. When inflated ($\Delta > 0$), the anisotropy of the actuator after being constrained is manifested as the actuator output bending moment and motion. Another is a tendon-driven flexible skeleton (Fig. 1c(ii)). The skeleton is manufactured with thermoplastic polyurethane (TPU) rubber by 3D printing (Supplementary Fig. 1d). Referring to our previous research³³, the skeleton opens as the tendon tightens and closes when it relaxes. Notably, only one tendon is required for a single skeleton, and three joints exhibit a sequential motion under the tension of one tendon when opening and closing due to stiffness gradients at these joints. In the finger closure process, the end joint is closed first, followed by the middle and fingertip joint. During opening, the fingertip joint is opened first, followed by the middle and end joint (Supplementary Movie 6).

The sequential motion behavior enables the skeleton to better wrap and is compatible with objects of various shapes compared to the uniform motion of the fiber-reinforced soft actuator mentioned above. However, the fact that the skeleton goes from flat to closed depends only on the internal stress of the material, which results in limited grip force and relatively long response time of the finger. For the proposed gripper, we present a solution that considers the kinematic and mechanical properties of both actuators, that is, inserting the fiber-reinforced actuator into the flexible skeleton to form a

composite (Fig. 1c(iii)). Thus, the grip force and response time of a finger are enhanced under a combined function of the air pressure inside the soft actuator and the material stress stored in the deformed skeleton. Coordinated by the FPSED mechanism (detailed below), the pre-charged gas flows between the fingers and palm. The tendons of the fingers tighten (F_{Tendon} increases), the internal actuator volume decreases as the fingers open, and gas flows into the palm (P_{Finger} decreases). The tendon is released when grasping an object (F_{Tendon} decreases), the object squeezes the palm while the finger is closed, and the gas flows into the actuator of the finger (P_{Finger} increases)."

Fig. 1 Overview of the proposed soft gripper and the FPSED mechanism. (a) A soft gripper mainly includes multiple soft fingers, an actuation module for finger tendons, a bellows-shaped flexible manipulator, and a control system. These fingers are actuated by tendons driven by a brushless DC motor in the actuation module. The fingers and palm are pre-inflated, and the gas is controlled by a valve system between the fingers and the palm. A manipulator is used to adjust the grasping direction of the gripper. (b) An actuation module consists of a motor and several gears. Gear 1 drives the surrounding planetary gears (gears 2, 3, and 4) and the winding discs on them to rotate to realize the retraction and release of the tendons. (i) The two-finger gripper has one central gear (gear 1) and two planetary gears (gear 2 and 3), (ii) while the three-finger gripper only needs to add one planetary gear (gear 4). (c) A soft finger is mainly composed of (i) a fiber-reinforced pneumatic soft actuator and (ii)

a 3D-printed tendon-driven skeleton. The fiber-reinforced actuator is inserted into the skeleton to build a soft finger whose grip and release are controlled by tendons and whose performance is enhanced by air pressure.

[R1] Zhang, Y., Zhang, W., Yang, J. L. & Pu, W. Bioinspired Soft Robotic Fingers with Sequential Motion Based on Tendon-Driven Mechanisms. *Soft Robot* 9, 531-541, doi:10.1089/soro.2021.0009 (2022).

2. To expand from the question above, based on Fig. 1a, there is "one" motor, but I don't think that is used to control the tendons? What exactly are in the control system? In Extended Data Fig. 1, what are the hardware components for muscle A and muscle B?

Response: Thank the reviewer for this question. As shown in the revised Fig. 1a-b mentioned above and in response to comment 1, the proposed soft gripper mainly includes a flexible manipulator, an actuation module for finger tendons, multiple soft fingers, and a soft palm. The function of the manipulator is only to adjust the position and orientation of the end gripper, and its control is independent of several fingers. For the manipulator, three tendons are arranged 120° along its circumferential direction and are controlled by three servo motors respectively. For the finger, their tendons are actuated simultaneously by a "motor" in the actuation module, which is achieved by a gear mechanism. In our application demo, we designed a two-finger gripper and a three-finger gripper. For the former, its gear transmission mechanism is shown in the revised Fig. 1b(i), and the motor drives a sun gear (gear 1) and two planetary gears (gear 2 and gear 3) meshing with the sun gear to rotate so that the winding disc rotates to realize the tightening and releasing of the finger tendons. For the latter, we just need to add a planetary gear and a winding disc around the sun gear, shown in the revised Fig. 1b(ii).

A control system is equivalent to a control box, in which three servo motors are used to drive the flexible manipulator, and some components are required for the FPSED mechanism, such as an Arduino microcontroller, solenoid valve, air pressure sensor, check valve, exhaust valve, battery, etc.

In the original manuscript, we used the working mechanism of human muscles to illustrate the

operating principle of the proposed finger. Muscle A represented the tendon and Muscle B represented the combined function of fiber-reinforced soft actuators and flexible skeletons. We appreciate your valuable questions. We have removed these unclear descriptions of Muscle A and Muscle B in the revised main text and redrawn the supplementary Fig. 1.

Supplementary Fig. 1. Design and fabrication of a finger and palm. (a) Both a fiber-reinforced bending actuator and a palm are made of silicone rubber material by casting. Their design parameters are shown in (b) and (c) respectively. (d) A flexible skeleton is made of thermoplastic polyurethane (TPU) by 3D printing technology based on Fused Deposition Modeling (FDM). (e) Front and isometric view of a flexible skeleton, including some design parameters such as joint bending angle, thickness, length, width, etc.

3. I am asking because, from my understanding, the palm and fingers are the two main components. But the fingers are not adequately described (I know that some may have been published and cited as previous works, but at the moment, the manuscript does not make sense in itself). And without this information, I am not yet convinced that the improved reaction time is critical in catching the objects.

Response: We thank the reviewer for this comment. We have supplemented the content of the working principle of the soft finger in the revised manuscript. Referring to the added content “**Components and working principle of the gripper**”, we aimed to clarify that, in the opening process, the driving force of a finger is provided by one motor-driven tendon. During the closure process, once the finger tendons are released, the fingers automatically complete the closure under the combined function of the stress stored in the flexible skeleton and the bending moment output by the fiber-reinforced soft actuator. The benefit of this design is that the higher the pressure inside the fiber-reinforced soft actuator, the higher the output bending moment, resulting in a faster finger response, which has been confirmed in our experiments (Fig. 3h).

Improved finger response time is an inherent property of the FPSSED mechanism and critical in capturing dynamic targets for a soft gripper. To capture fast dynamic objects, it is key that the gripper has the capability to overcome the "fast" of dynamic objects. As shown in Fig. 3f and supplementary Fig. 6a-d, the collision time is extremely short between the ball and the palm from contact to complete disengagement, which requires the gripper to respond quickly enough to capture. Actually, it is very difficult for most soft grippers to meet such high responsiveness requirements.

In our study, on the one hand, we exploit the large deformation properties of the soft palm to prolong the compression time, as shown in Fig. 3f. On the other hand, we use the FPSSED mechanism to quickly dissipate the object's kinetic energy and eliminate their bounce after a collision. This makes the collision time more than 10 times longer than the milliseconds (4 ms) of a collision between rigid bodies. From the experimental results as shown in Fig. 3h, although the increased response time of the finger is only tens of milliseconds, it accounts for about 30% of the total collision time. After prolonging the collision time and eliminating the bounce, the effect of gravity is not negligible in real applications, so it is very important that the gripper quickly wraps the object to prevent it from slipping in the direction of gravity.

4. Following above, a palm designed with Design C (completely opened) seems to do quite well for dampening out the energy. With the leftover energy of less than 0.1 J, the glass ball bouncing speed is around 0.8 m/s(?), it would travel around 3 cm in 40 ms, would that be sufficient time for normal

fingers to catch the ball? I cannot answer because I do not understand how the fingers are activated.

Response: Thanks for your valuable question. We guess that the case you mentioned should be Design D. As described in the **Introduction**, soft grippers are mostly made of low-modulus elastic materials. Their ability to resist slight dynamic interference through large deformation is better than that of rigid grippers. Generally, objects with lower kinetic energy or speed can also be grasped by some soft grippers. However, because of the low modulus of the material, the load capacity of the gripper is usually weak. If the rebound kinetic energy of a target is too large, the target can easily break through the maximum bearing capacity and escape. As verified in our experiment, the energy dissipation performance of these typical design palms (Design A, B, C, and D) is difficult to regulate and has poor adaptability (Supplementary Fig. 6). For them, the faster the target object is, the more difficult it is to stop it.

We proposed to design the FPSED mechanism on a soft gripper to make a dynamic target stop completely after a collision, which is the desired effect for a soft gripper. More importantly, the compatibility and scalability of palm performance are also considered. The palm can adapt to the target object at various speeds only by adjusting the pre-charged pressure. It also improves the grasping force and response time of fingers to achieve robust capture.

Supplementary Fig. 6. Comparison of the collision results of four typical palms. (a) Rigid material. The palm is made of three-dimensional (3D) printed polylactic acid (PLA). (b) Solid rubber pad. This palm is made of silicone rubber with a Shore hardness of 5 and is completely solid. (c) Completely blocked. The palm is an airbag made of silicone rubber, and its two exhaust ports are completely blocked. (d) Completely opened. The material and design are the same as (c), except that its two exhaust ports are completely connected to the atmosphere. Each set of graphs represents the kinetic energy dissipation of the ball during a collision, and the color map represents the velocity decay of the ball. The ball hits the palm from 20 ms.

5. Another issue is the focus. Based on the introduction, the focus of the work is on damping out the KE for grasping. Suddenly, from Lines ~250, the consideration on releasing (ungrasp) and staility of a grasp is brought up without any context. I also do not see a tight connection with the main contribution of the work. I find it confusing and even distracting, let alone the fact that is difficult to follow or deduce the message the authors try to convey here.

Response: We thank the reviewer for this question. Referring to your suggestions, we have re-adjusted the description of the FPSED mechanism and modified the schematic diagram as shown in the revised Fig 1d.

The proposed FPSED mechanism mainly addresses the energy dissipation during a collision. The original Fig. 1b(iii) (Releasing target) is actually the action after a collision and capturing have been completed, and it is less proper to describe it in the collision process. In the revised manuscript, we divided the key process of the FPSED mechanism into three stages: before collision, during collision, and capturing target. Before collision, the initial pressure in the palm and fingers is the same. (Fig. 1d(i)). During collision, the kinetic energy of an object will be converted into gas internal energy to stiffen the gripper and improve gripping, and the gas can only flow in one direction (palm to fingers) (Fig. 1d(ii)). When capturing a target, the control board quickly opens the exhaust valve. In this way, the remaining energy stored in the palm is dissipated following rapid decompression(Fig. 1d(iii)).

Fig. 1 Overview of the proposed soft gripper and the FPSSED mechanism. (d) Three essential stages of the FPSSED mechanism: (i) before collision, (ii) during collision and (iii) capturing target. (e) A six-rotor drone integrated with the proposed soft gripper is presented and demonstrated its ability to capture dynamic targets outdoors. Scale bars represent 0.2 m.

===== Other Major points =====

1. The use of the "magnet" analogy is not scientific nor useful. The gripper does not ACT like a magnet as it does not attract the object. Energy dissipation and harvesting are also not properties of magnetic forces.

Response: We thank the reviewer for the great suggestions. We fully agree with the reviewer and have removed the description of "magnet" from the manuscript for a more rigorous and objective presentation.

0. In the submitted version, there is no evident separation between the abstract and the main text. There are no section titles (Introduction/Results/Discussion). The separation is also unclear in the content of the main text. There also exists no headings for subsections in the Result "section". This makes it rather difficult to follow the manuscript. In fact, I tried to figure out where the transition from "Introduction" to "Results" is, but I cannot really pinpoint where.

Response: We thank the reviewer for the comment. This is indeed our fault on the format issue as the paper was redirected from *Nature* main magazine. We have divided chapters and subsections in the revised manuscript for clarification. The current organization is aligned with the requirements of *Nature Communications*.

1. Line 192: "can reach 100%". I will be a little more careful with the claim as it sounds like it is always 100%. You might want to add "can reach 100% when ...", or make the claim less bold as "could/may reach up to 100%".

Response: Thank you for your constructive suggestion! We have revised the argument to be more rigorous in the **Abstract** and **Results**.

2. Lines 196-200: How are "harvesting efficiencies" and "elastic potential" energy defined and measured?

Response: Many thanks for this question. We have divided a subsection "Energy dissipation capability" and supplemented definitions and measurement methods of "harvesting efficiency" in the

revised manuscript. Furthermore, the formulation of "elastic potential energy" in the previous manuscript was not accurate enough and we have re-described it.

We define the energy harvesting efficiency η_h as the percentage of the harvested energy E_h to

the initial total kinetic energy K_i of a ball ($\eta_h = \frac{E_h}{K_i} \times 100$). More details are added in the main text,

as follows:

"Energy dissipation capability. The FPSED mechanism theoretically manifests a superior energy dissipation capacity to avoid rebound. We also observed experimentally that the energy of a ball can be almost dissipated entirely, and the experimental results are shown in Fig. 3c-3e and Supplementary Video 2. In the FPSED mechanism, there are two designed energy dissipation pathways, one is to transfer part of the energy to the finger, and the other is to dissipate by releasing gas. In addition, the rest of the kinetic energy of the ball is dissipated by compressive deformation of the elastic palm, system vibration, friction, etc. We define the energy harvesting efficiency η_h as the percentage of the

harvested energy E_h to the initial total kinetic energy K_i of a ball ($\eta_h = \frac{E_h}{K_i} \times 100$). The energy dissipated E_d by releasing gas as a percentage of the total energy is defined as the energy dissipation

efficiency η_d ($\eta_d = \frac{E_d}{K_i} \times 100$). Further, outside these two parts, dissipating the residual kinetic energy of a ball depends on the material and structure of a palm as well as vibration and friction, which belong to the inherent energy dissipation properties of a palm.

To measure the E_h and E_d , we first collect a set of kinetic energy data based on the FPSED mechanism through collision experiments (Fig. 3c(i)-3e(i)). Then, with the exhaust valve always closed, we obtain a set of kinetic energy data without energy harvesting (Supplementary Fig. 5). Finally, with both the exhaust and gas transfer pipes closed, we collect a set of kinetic energy data without energy harvesting and air release (Supplementary Fig. 6c). For these three sets of data, the remaining kinetic energy is extracted when the ball is separated from the palm, which is recorded as K_{r1} , K_{r2} , and K_{r3} . Because the experiential results of collisions with three different initial kinetic energies show that $K_{r1} = 0$, E_h and E_d can be obtained by $E_h = K_{r3} - K_{r2}$ and $E_d = K_{r2}$. Quantitatively, for the FPSED mechanism, the

experimental results with three different initial kinetic energies (0.14 J (Fig. 3c(i)), 0.57 J (Fig. 3d(i)), and 0.88 J (Fig. 3e(i))) demonstrate that their energy harvesting

efficiencies η_h are 27.86%, 29.82% and 31.82%. Their energy dissipation efficiencies η_d are 41.43%, 31.58% and 26.13%, respectively. Noticeably, these results perfectly agree with the simulation results computed by the model we built above. These results illustrate that the greater the kinetic energy of the incoming target, the greater the energy harvested by the FPSED mechanism. This result reduces the energy that needs to be dissipated by deflation. Under the relatively large kinetic energy impact, we observed that the increased deformation of the palm increases the proportion of energy dissipated by the palm deformation. Importantly, benefiting from the FPSED mechanism, the initial total energy is eventually dissipated.”

5. Pre-loading the palm with positive pressure allows an object with higher KE to be stopped completely

Is there a notable disadvantage of using high positive pressure in all cases (apart from lowering the harvesting efficiency)? Regardless of the answer, can you provide supporting data/measurements?

(I guess this is Extended Data Fig. 6, but it would be great to summarize this point in the main text)

Response: Thanks for your valuable question. With reference to your good suggestion, we have supplemented some experiments and re-summarized our views in the revised **Results** section “**Performance of the FPSED mechanism**” and “**Compatibility and scalability**”. Our response to this comment is as follows:

In all cases, inflating high positive pressure in a palm mainly affects the capture effect for projectiles with relatively low kinetic energy. In this study, the energy harvesting efficiency η_h is a comprehensive evaluation index, which affects the response time and grasping force of a finger on the one hand, and the total energy dissipation efficiency η_d on the other hand.

On the basis of comparing the energy harvesting efficiency, we systematically compared the effect on the finger grasping force and response time when using the high-pressure palm to capture low-speed objects. It can be found that not only the harvesting efficiency will be significantly reduced, but also the grasping force of fingers will significantly decrease, and the finger response speed will also be significantly slowed, which is not conducive to the capture of dynamic objects. As shown in Fig. 3g, under the impact of the ball with a kinetic energy of 0.88 J, the pressure increment inside the

syringe is increased by 14.5 kPa when the palm is inflated to 30 kPa. In contrast, when the kinetic energy of collision decreases to 0.14 J and the palm pressure is maintained at 30 kPa, the pressure increment in the syringe drops sharply to about 0.2 kPa, which is about 1/70 times of the former. The energy harvesting efficiency is reflected in the pressure increment, which directly affects the grasping force and response time of the fingers. Taking the two groups of pressure inside the finger 0 kPa and 30 kPa for comparison, the difference in grasping force and response time is shown in Fig. 3h. At 0 kPa, the average grasping force is 16.99N, which is lower than 26.28N at 30 kPa. In addition, the finger response time is 48 ms, which is slower than 34 ms at 30 kPa.

In the FPSED mechanism, we expect more energy to be harvested rather than dissipated by releasing gas. If a target with relatively small kinetic energy is captured with a high-pressure palm, the energy harvesting efficiency will be reduced, which will cause more energy to be dissipated by releasing gas. Theoretically, this system based on the FPSED mechanism can adaptively maintain the balance between energy harvesting and dissipation according to the target kinetic energy and the pressure control, so that objects of various speeds do not rebound after a collision. In the practice, the commercial exhaust valve has a limited vent size and response time, with high positive pressure in the palm, if the collision speed is relatively low, the proportion of energy to be released by releasing gas can be very high, we have to adopt an exhaust valve with a larger vent size and faster response speed. We supplement a set of experiments and add the data of the ratio of the energy dissipated by releasing to the initial total energy under different pre-charged pressures and collision velocities. As shown in Supplementary Fig. 7, for a low kinetic energy collision (0.14 J), if the palm is inflated with additional pressure (from 10 kPa to 30 kPa), the energy that needs to be dissipated by releasing gas accounts for more than 60% of the total energy. In particular, the proportion at 30 kPa exceeds 80%.

Supplementary Fig. 7. Relationship between the initial collision kinetic energy and the energy dissipation efficiency required for the FPSED mechanism under different pre-charged pressures.

6. I do not entirely agree with this sentence "We can assume that the collision duration of the FPSED mechanism is infinite" (Line 234). The collision time is until the object comes to a complete stop.

Response: Thank you for your question. We fully agree with you that the collision duration cannot be infinite. We originally wanted to express that the object is not bouncing and the ball is always in contact with the palm. Indeed, such a statement is not scientific, and we have rewritten it.

2. The paragraph from Line 248 to 278 should be re-written. It starts with ".. to evaluate the effect of the air pressure transmitted by the collision on the blocking force and response time." But after talking about the blocking force, it transitions to "slipping and stable grasping" instead of talking about the response time (which eventually comes later). The content on stable grasping is also a little out of context as mentioned earlier.

Response: Many thanks for your valuable comment. "slipping and stable grasping" is used to describe the contact relationship between the object and the finger during the pulling process from being grasped to completely detached from the finger. To more clearly, "blocking force" has been replaced by "grasping force". "stable" has been replaced by "steady". To make this section easier to understand, following your suggestion, we have rewritten the "**Effect of the harvested energy on fingers**" section in the revised manuscript as follows:

“Effect of harvested energy on fingers. After solving how to stop a dynamic target quickly and maintain a longer contact time, the gripper possesses the key and primary conditions for grasping a dynamic target. Next, we evaluate how the energy harvested during a collision improves the grasping force and response time of a finger.

First, we analyze how much the air pressure in a closed cavity is raised in a collision by collecting the data recorded through the pressure sensor in the collision experiment above, and plot the relationship of pressure increments inside the syringe at different pre-inflation pressures corresponding to three initial kinetic energies of 0.14 J, 0.57 J, and 0.88 J (Fig. 3g). These results indicate that the larger the collision kinetic energy of the target, the more gas will be transferred when the pre-charged pressure is constant. When the pre-charged pressure increases, the gas transfer capacity decreases. For example, when the pre-charged pressure is 0 kPa, the pressure increment corresponding to the collision kinetic energy of 0.14 J is close to 7 kPa. When the kinetic energy increases to 0.88 J, the pressure increment is significantly increased to 26 kPa. When the kinetic energy increases to 0.88 J, the pressure increment is significantly increased to 26 kPa. In contrast, when the pre-charged pressure is 30 kPa, the pressure increment drops to 15 kPa.

Then, based on the above results, we separately input air pressures of 0-30 kPa in increments of 10 kPa into the bending actuator to evaluate the effect of the air pressure transmitted by the collision on the grasping force. We apply a pulling force to pull a 60 mm diameter cylinder with negligible mass from the fully wrapped state, and then use the variation of this pulling force over time to illustrate the grasping ability of a finger (Supplementary Fig. 9a). Through the continuous grasping force curve (Supplementary Fig. 9b-e), we find that the process of an object detaching from a finger follows a general rule: the object needs to go through three stages steady grasping, slipping, and sudden detachment before it can completely escape from the finger. Referring to our previous research³³, we certified that the stiffness gradient on the finger is a critical reason for the evolution of the grasping force that first increases and then slowly decreases and finally disengages. The end joint and middle joint are essential in resisting object displacement in the stable grasping stage. However, with the further displacement increase, the object quickly slipped from the high-stiffness end joint and middle joint to the low-stiffness middle and fingertip joint. Especially in the early stage of complete

detachment, only the fingertip is left to function. In the stable grasping stage, the grasping force gradually increases while the target object moves, and the target surface is in close contact with the finger surface. After starting to slip, the grasping force gradually decreases until the maximum detachment force is reached, during which the target rolls on the inner surface of the finger. As shown in Fig. 3g, without additional inflation, the object transitioned from a steady to a non-steady state starting at 23.2 N, and completely disengaged from the fingertip when the force dropped to 14.1 N. In contrast, when the pressure in the bending actuator rises to 30 kPa, the object starts to slip at 30.9 N, and a more significant force of 26.6 N is required to drag the object to disengage it. The average forces in the slip area at the air pressure of 0 kPa, 10 kPa, 20 kPa, and 30 KPa are 17.0 N, 19.5 N, 22.6 N, and 26.3 N, respectively.

Eventually, we characterize that the harvested energy accelerated the finger's response time while improving the grasping force. To capture fast dynamic objects, it is key that the gripper has the ability to overcome the "fast" of dynamic objects. As shown in Fig. 3f and supplementary Fig. 6a-d, the collision between the ball and the palm from contact to complete disengagement, is extremely short, which requires the gripper to have a fast response time to capture. In fact, it is very difficult for the reported soft gripper to meet such high responsiveness requirements. Here, we employ a high-speed camera (1000 fps) to measure the time it takes for a finger to be released from a fully flattened state until the finger just wraps around a 60 mm diameter cylinder. Experimental results in Fig.3h show that with no additional air pressure in the fingers, it took 48 ms from the system to go transit from fully flattening to grabbing a 60 mm diameter cylinder. As the pressure inside the finger increases, the time corresponding to the same behavior is significantly shortened. For example, when the pressure is 30 KPa, the response time is improved by 28%, and it only takes 34 ms to capture the object.

Taken together, objects with more kinetic energy tend to require greater grasping force and response time from the fingers. In this study, benefiting from the FPSED mechanism, the finger grasping force and response time are positively related to the kinetic energy of the objects, which greatly reduces the difficulty of control and improves the grasping stability.”

8. Line 264: "As shown in Fig. 3g, ..." I do not see what is described here in Fig. 3g.

Response: We are sorry for the inconvenience caused to you. Due to our negligence, "As shown in Fig. 3g,..." is actually "As shown in Fig. 3h,". We have corrected these errors in the revised manuscript.

9. In the Conclusion, starting from line 332, I agree with "A highlight of the FPSED mechanism is that its capabilities are adjustable and extendable simply by setting the pre-inflating pressure,...". However, other points, such as "protecting targets and themselves", and "These properties will be important for robotic grasping in highly cluttered environments where collisions are difficult to avoid." are not clear and not demonstrated in this work. I do not believe you can make these conclusions.

Response: Many thanks for your comment. One of the most significant advantages of soft robots over rigid robots is interaction safety, as they are usually made of soft elastic materials with low Young's modulus [R2]. Therefore, from a macroscopic perspective, an object collides with a soft material with a large deformation capacity, and it is undoubtedly more dangerous and damaging to collide with a rigid material than with a rigid material. Indirectly, although there is no direct data to prove that in this study, a soft palm made of silicone rubber material and fingers made of soft polyurethane material can protect the target and itself more than steel and aluminum grippers.

Based on your comments, we also believe that it may not be very reasonable to put it in the conclusion at this time. Therefore, we have revised the unreasonable conclusions in the **Discussion** section more rigorously, including your mention of "These properties.....in highly cluttered environments are difficult to avoid.". In the future, we will continue to work on these important issues and provide scientific evidence to prove these views.

[R2] Rus, D. L. & Tolley, M. T. Design, fabrication and control of soft robots. Nature 521, 467-475 (2015).

10. Line 336 "we added a 3D printed tendon-driven bellows-like flexible manipulator to the gripper." Is this bellow (actuated) structure solely for positioning/re-orienting the gripper? Please make it clear that it is an entirely separate component.

Response: Thanks for your question. In the application demonstration, we set up a bellows-shaped manipulator on the base of the gripper to adjust its position and orientation, which is more accurate and convenient than directly adjusting the height and steering of the drone. Therefore, it has been clearly described in the " **Components and working principle of the gripper**" section that the manipulator is an entirely separate component.

0. Line 417-418 refers to air inlet A and air output P, but which figure are you referring to?

Response: Thanks for your question. We have marked in revised Fig. 1 the letter labels corresponding to each interface of the solenoid valve.

1. Dynamics model: lots of variables undefined. P is pressure? How does one calculate f_{corre} ?

Response: Many thanks for your question. We further completed the model and explained in detail the meaning of each parameter in the revised manuscript.

REVIEWERS' COMMENTS

Reviewer #3 (Remarks to the Author):

I would like to thank the authors for the revision effort. It is evident that the authors have majorly improved the manuscript based on the comments I provided. The description and working principles of the device are now much clearer. I do not have any remaining major concerns.

If applicable, for the final submission or subsequent updates, I strongly recommend the authors to

1) consider having the manuscript polished. There remain sentences that are grammatically awkward, particularly the revised text. This is brought up as it sometimes affects the meaning. The revision would benefit the flow of the paper.

2) add a few sentences to justify the choice of the 60-mm 283-g glass ball used for most experiments. Why would this object be a sufficient representation of a real-world object? In which cases it is not? This limitation of the work and the results (as a consequence of the decision to use this ball for most experiments) should be adequately mentioned.

Responses to the reviewers' comments (NCOMMS-22-26383A)

REVIEWERS' COMMENTS

Reviewer #3 (Remarks to the Author):

I would like to thank the authors for the revision effort. It is evident that the authors have majorly improved the manuscript based on the comments I provided. The description and working principles of the device are now much clearer. I do not have any remaining major concerns.

Response: We greatly appreciate your review and valuable suggestions. With reference to your comments, we have again perfected our manuscript.

If applicable, for the final submission or subsequent updates, I strongly recommend the authors to 1) consider having the manuscript polished. There remain sentences that are grammatically awkward, particularly the revised text. This is brought up as it sometimes affects the meaning. The revision would benefit the flow of the paper.

Response: Many thanks for your suggestions. We have carefully checked the full text and improved the expression and grammar. All modifications have been highlighted in the main text.

2) add a few sentences to justify the choice of the 60-mm 283-g glass ball used for most experiments. Why would this object be a sufficient representation of a real-world object? In which cases it is not? This limitation of the work and the results (as a consequence of the decision to use this ball for most experiments) should be adequately mentioned.

Response: Thank you for your valuable suggestions. We have added some sentences in the main text to explain why we chose a 60-mm and 283-g glass ball for most experiments. Our research aims to capture a target with kinetic energy by a new soft gripper. The target's kinetic energy is determined by its speed and mass. Changing the speed or mass of the object can obtain different kinetic energy. Here, we chose a relatively simple approach that altering the object's velocity to acquire various kinetic energy for quantitative comparison.